# Effectiveness of Adding a Pain Neuroscience Education Program to a Multimodal Physiotherapy Intervention in Patients with Chronic Shoulder Pain: A Randomized Clinical Trial

**DOI:** 10.3390/brainsci15020125

**Published:** 2025-01-27

**Authors:** Jose Angel Delgado-Gil, Eva Prado-Robles, María Nieves Muñoz-Alcaraz, Jesús Seco-Calvo

**Affiliations:** 1Primary Care Service Area, Castilla y León Health Service, 24008 León, Spain; 2Unit of Physical Medicine and Rehabilitation, León University Hospital, Castilla y León Health Service, 24008 León, Spain; eprado@saludcastillayleon.es; 3Unit of Physical Medicine and Rehabilitation, Reina Sofía University Hospital, 14004 Córdoba, Spain; marian.munoz.sspa@juntadeandalucia.es; 4Maimonides Biomedical Research Institute of Córdoba (IMIBIC), Reina Sofía University Hospital, University of Córdoba, 14071 Córdoba, Spain; 5Institute of Biomedicine (IBIOMED), León University, 24071 León, Spain; jesus.seco@unileon.es

**Keywords:** pain neuroscience education, shoulder pain, manual therapy, exercise

## Abstract

**Objectives**: The purpose of this study was to assess the efficacy of a pain neuroscience education program completed by patients with shoulder pain. **Methods**: A randomized, controlled trial was performed. Fifty-five patients satisfied the eligibility criteria, agreed to participate, and were randomized into an experimental group (n = 27) or control group (n = 28). A manual therapy and exercises program was administered for both groups. The experimental group also received a 4-week pain neuroscience education protocol (1 session/week, 75 min per session). The measurements taken included the active range of motion, pain, disability, catastrophizing, kinesiophobia, and therapeutic alliance. The outcomes were assessed at baseline and 5 weeks after completion of treatment. The primary outcome analyzed was the group × time interaction. **Results**: The 2 × 2 analysis of variance revealed a significant group × time interaction for the active range of motion (F = 15.27; *p* = 0.011), disability (F = 6.14; *p* = 0.01), catastrophizing (F = 8.79; *p* = 0.01), kinesiophobia (F = 7.62; *p* = 0.008), and therapeutic alliance (*p* = 0.03) in favor of the experimental group. **Conclusions**: This study showed that the patients with shoulder pain who completed the pain neuroscience program achieved significantly better results in terms of their active range of motion, disability, catastrophizing, kinesiophobia, and therapeutic alliance compared to those achieved by the control group. Therefore, pain neuroscience education may be beneficial in the treatment of patients with shoulder pain.

## 1. Introduction

Shoulder pain (SP) is the third most common musculoskeletal disorder, with an incidence from 7.7 to 62.0 per 1000 persons per year, and a point prevalence from 0.7 to 55.2% in the general population [1]. The recovery from shoulder pain can be slow and recurrence rates are high, with 25% of those affected by shoulder pain reporting previous episodes and 40–50% reporting persistent or recurrent pain at a 12-months follow-up [2].

This prognosis may be influenced by age, a low educational level, a long duration of symptoms, previous episodes, high rates of disability, pain in other body areas [3], and the presence of psychosocial factors [4]. These psychosocial factors may be risk factors and triggers for the onset and development of shoulder pain, may be facilitators of or barriers to patient recovery [5], and appear to play a key role in explaining why musculoskeletal pain becomes chronic after the normal period of tissue healing has elapsed [5]. The presence of these psychosocial factors may suggest that a pain neuroscience education (PNE) approach could be considered as part of the treatment for these patients, as there is a growing body of research on its efficacy and effectiveness for different pathologies [6].

PNE is an educational strategy that focuses on teaching people in pain more about the neurobiological and neurophysiological processes involved in their pain experience, especially for chronic pain, to influence their beliefs and behaviors [7]. PNE could have a positive effect on pain, disability, catastrophizing, and kinesiophobia [8]. However, the effectiveness of PNE as an isolated treatment technique is limited and on its own may not help to improve the impaired movement and muscle performance observed in individuals with shoulder pain [9]. Thus, exercises (EX) [10] and manual therapy (MT) [11] may be indicated to address such impairments, as they produce a number of beneficial physiological and biomechanical effects [12]. MT is used to reduce pain and increase joint mobility, and EX aims to improve muscle function and range of motion by restoring mobility, proprioception, and stability to the shoulder [13]. When used together, they are considered effective in the treatment of certain shoulder conditions with severe pain, such as rotator cuff disease [14] or adhesive capsulitis [15,16].

Moreover, manual therapy is based on Mulligan’s movement mobilization [17]. Mulligan’s mobilization with movement (MWM) technique is a manual therapy approach involving the application of a sustained gliding force (passive mobilization component) with a concurrent active movement or a functional task performed by the patient (active movement component) [18]. The application of MWM, when precisely indicated, has beneficial effects on painful movements, and, thereby, function is immediately improved [19].

In this regard, exercise therapy for shoulder pain is effective and consistently recommended, but in general, there is no consensus on the type, intensity, frequency, or duration of the exercises used [20]. Some of the most recommended types of exercises are strengthening exercises, as they have been shown to be effective in reducing pain and disability in people with shoulder pain [21].

Despite the growing evidence supporting the effectiveness of PNE in conditions such as chronic spinal pain [22,23], there is a lack of studies evaluating its applicability and effectiveness in patients with chronic shoulder pain [24]. While the relationship between psychosocial factors and shoulder pain is documented in the scientific literature [25], studies that specifically apply this treatment approach to shoulder pain are scarce. Our study aims to fill this gap by evaluating the effectiveness of PNE in patients with chronic shoulder pain, expanding the usual conservative treatment for painful shoulders, which typically includes rest, non-steroidal anti-inflammatory drugs (NSAIDs), corticosteroid injections, and physiotherapy treatments based on exercises and various manual therapy modalities [26].

In view of this, the aim of this randomized clinical trial is to compare the effects of two interventions for people diagnosed with shoulder pain: MT and EX, and PNE, MT, and EX. The primary aim of the study was to compare pain intensity, disability, and mobility between the experimental and control groups. Secondarily, we also investigated the comparison of scores of catastrophizing, kinesiophobia, and satisfaction with the working alliance. We hypothesized that the addition of a PNE program for exercise and manual therapy could improve outcomes in patients with chronic shoulder pain.

## 2. Materials and Methods

### 2.1. Study Design

A repeated-measures, single-blind, randomized controlled trial was conducted following the CONSORT guidelines. The study protocol was approved by the Ethics Committee of León University Hospital, Spain, and its internal registry number was 21179. This research was conducted in accordance with the World Medical Association Declaration of Helsinki. Patients were informed about the study and provided their written consent, and the study was registered at ClinicalTrials.gov with the identifier NCT06739694.

### 2.2. Participants

The study population consisted of 56 patients with shoulder pain selected from the waiting list for Primary Care in León (Castilla y León Public Health Service, SACYL).

Patients were included if they had the following: (1): age 18–70; (2) history of shoulder pain of more than 3 months’ duration; (3) presence of a painful arc; (4) medical diagnosis of shoulder pain with at least 2 positive impingement tests, including Neer, Hawkins, or Jobe tests [27]. The exclusion criteria included (1) a diagnosis of fibromyalgia; (2) pregnancy; (3) a history of traumatic onset of shoulder pain; (4) other histories of shoulder injury; (5) torn tendons; (6) ligamentous laxity based on a positive sulcus and apprehension tests; (7) numbness or tingling in the upper extremity; (8) previous shoulder or cervical spine surgery; (9) systemic illness; (10) corticosteroid injection in the shoulder within 1 year of the study; and (11) physical therapy 6 months before the study.

### 2.3. Randomization and Blinding

Concealed allocation was performed using a computer-generated randomized table of numbers (Random.org) created before the start of data collection by a researcher not involved in the recruitment and/or treatment of the patients. Individual and sequentially numbered index cards with random assignment were prepared. The index cards were folded and placed in sealed opaque envelopes. A researcher blinded to group assignment conducted the assessments at baseline and 5 weeks after the intervention, on an individual basis and without matching patients to each other or to the therapist who conducted the treatment to prevent interactions or information exchange. A second therapist, blinded to baseline examination findings, opened the envelope and proceeded with treatment according to the group assignment. The participants were randomly assigned to either the experimental group (PNE + MT + EX) or the control group (MT + EX).

Due to the type of study carried out, it was not possible to blind the physiotherapist in charge of the interventions, which may have been a limitation of the study. Participants were unaware of the purpose of this study, and were informed that the study was a comparison between two physiotherapy treatments. Patients had no prior experience or knowledge of PNE, and there was no overlap between subjects in the experimental and control groups, which contributed to blinding. The statistical analysis was also performed by a blinded researcher who had no knowledge of the groups to which each patient be-longed.

### 2.4. Interventions

A physiotherapist with 20 years of experience administered the treatment program. For the control group, the MT and EX program lasted five weeks with one session per week, while in the experimental group, the program started four weeks earlier with one session per week of PNE, followed by the addition of MT and EX therapy (Figure 1). Measurements were taken prior to the start of treatment and five weeks after the end of manual therapy and exercises.

#### 2.4.1. Manual Therapy

Certain manual therapy techniques, such as Maitland [28] or Kaltenborn [29], have proven effective in treating shoulder pain. However, for this study, we selected the Mulligan mobilization with movement technique not only for its effectiveness [30] but also because it is frequently used in our workplace environment. Although used as a treatment technique, MWM is also considered an assessment procedure [31]. The starting point is to select a movement or activity that reproduces the patient’s symptoms. Once the provocative activity is known, one or more techniques that reduce the symptoms, either by reducing pain and/or increasing movement, are identified [32]. Based on the information gathered during the clinical interview and physical examination, a decision is made as to which joint to treat and the direction of the physiotherapist’s performance of the passive technique [33]. Exploratory and/or treatment techniques target the humeral head [34], scapula [35], and cervical [36] and thoracic regions [37]. During the sessions, passive accessory gliding was applied while the patient performed the active movement or function that reproduced their symptoms. Each exercise was performed in three sets of 10 repetitions, with a total duration of 15–30 min [38].

#### 2.4.2. Exercises

Participants were instructed to perform a home-based program of progressive shoulder-strengthening exercises [39] involving concentric and eccentric contractions with elastic resistance bands. The exercises targeted the humeral internal/external rotators, abductors, and scapular muscles [40]. The exercises were performed in three sets of 10 repetitions, four times per week, with a 1 min interval between repetitions during the first week. From the second week onwards, the exercises were increased to 12 repetitions, and from the third week onwards, they were increased to 15 repetitions.

#### 2.4.3. Pain Neuroscience Education

In addition to the MT and EX treatment, a PNE program was added for the experimental group for a period of 4 weeks (1 session/week, 75 min per session, 5/6 patients per group) by a physiotherapist with 6 years of experience. PNE aimed to re-conceptualize pain perception from a biomedical or structural model to a biopsychosocial pain model through education on the neurophysiological aspects of pain [41]. All sessions used a slide presentation (PowerPoint, Microsoft Corp., Redmond, WA, USA) prepared by the investigator based on previous protocols and modified to treat shoulder pain [42,43]. The full content of the education is shown in Table 1.

### 2.5. Outcome Measures

The study participants were assessed at baseline and at the end of the interventions by a physiotherapist who was not involved in the study and who was blinded to the treatment performed. Prior to the start of the study, a series of socio-demographic variables were collected: age, sex, dominance, location of pain, and duration of symptoms (months).

#### 2.5.1. Shoulder Function

The primary outcome measure was the Shoulder Pain and Disability Index (SPADI). The SPADI is a self-reported questionnaire that evaluates the pain and disability associated with shoulder diseases [44]. This questionnaire comprises 2 subclasses (pain and disability) with 13 items (5 items in the pain domain and 8 items in the disabilities domain). The SPADI is scored between 0 and 100, and the SPADI score is calculated by averaging the scores from the 2 subclasses. A higher SPADI score indicates more severe symptoms and a greater level of disability. The questionnaire has been validated for use in Spain [45]. In patients with shoulder pain, the minimal detectable change (MDC) for the SPADI is reported to be 18.1, and the minimum clinically important difference (MCID) has been established as 13.2 [46].

#### 2.5.2. Shoulder Flexion Active Range of Motion (AROM)

Shoulder flexion active range of motion (AROM) was measured according to international guidelines using a goniometer [47]. AROM measurement using a goniometer has excellent intra-inspector reliability (ICC = 0.91–0.99) [48]. Despite the increasing use of inertial measurement unit (IMU) systems in shoulder joint measurements, only a few studies have tested the validity of IMUs compared to goniometers [49], which is why we have chosen to use them. The minimum detectable change (MDC) for shoulder flexion has been reported as 8°, and the calculation of the minimum clinically important difference (MCID) depends on the patient’s pathology, but it is generally accepted that a change of 6° to 11° is needed to be certain that a true change has occurred with goniometric shoulder measurements [50].

#### 2.5.3. Catastrophizing

The Spanish version of the Pain Catastrophizing Scale (PCS) will be used [51]. This is a brief 13-item questionnaire that assesses pain-related behaviors and cognition. Scores range from 0 to 52, with higher scores indicating a higher level of catastrophizing [52]. In patients with shoulder pain, the MDC for the PCS is reported to be 7.97 [53].

#### 2.5.4. Kinesiophobia

The Spanish version of the Tampa Scale of Kinesiophobia (TSK-11) was used [54]. This is a self-reported questionnaire containing 11 items designed to assess a patient’s fear of moving and reinjury. Scores range between 11 and 44 points, with a higher score indicating higher levels of kinesiophobia. The MCD for this tool in patients with shoulder pain has been reported to be 5.6 points [55].

#### 2.5.5. Therapeutic Alliance

Therapeutic alliance was measured with the Working Alliance Inventory (WAI) [56]. The WAI is a measure of therapeutic alliance that assesses three aspects: agreement on therapy tasks, agreement on therapy goals, and bonding. It consists of 36 items with scores ranging from 36 to 252. Higher scores indicate a higher level of therapeutic alliance.

### 2.6. Sample Size

The sample estimation was calculated through a previous feasibility study [57] with SPADI score as primary outcome measure and using G*Power Software (Version 3.1.9.7, Düsseldorf University, Düsseldorf, Germany). The minimum required sample size was calculated as 50 participants for the anticipated effect size of 0.82 with a probability level of 0.05 and statistical power level of 80%. Considering a dropout rate of 10%, 55 participants were recruited.

### 2.7. Statistical Analysis

All statistical analyses were performed using IBM SPSS Statistics version 27.0 (IBM Corp., Armonk, NY, USA). Descriptive statistics were used to summarize baseline data, including means and standard deviations for continuous variables and numbers and percentages for categorical variables. A Kolmogorov–Smirnov test showed a normal distribution of all quantitative data (*p* > 0.05). Continuous variables were compared using a *t*-test, while categorical variables were compared via a chi-squared test. An independent *t*-test was performed to determine when the interaction between groups appeared over time. A 2 × 2 repeated-measures analysis of variance (ANOVA) with time (pre-treatment vs. post-treatment) as the within-subjects factor and the group (experimental vs. control group) as the between-subjects factor was used to determine the effects of the intervention. When an interaction was detected, a post hoc test was performed using a Bonferroni test. The partial eta square was calculated to classify effect size (η*p*^2^) [58]. The statistical analysis was conducted at the 95% confidence level, and *p* < 0.05 was considered statistically significant.

## 3. Results

Sixty-seven (n = 67) consecutive patients with shoulder pain were screened using the eligibility criteria. Fifty-five patients satisfied the eligibility criteria, agreed to participate, and were randomized into the experimental group (n = 27; mean ± SD age, 57.22 ± 1.57) or the control group (n = 28; mean ± SD age, 56.18 ± 1.71). A study flowchart is presented in Figure 2.

All of the demographics were similar between the groups (Table 2). There was no difference between the groups in terms of the baseline clinical outcomes (*p* > 0.05).

Comparisons of the outcome measurements within and between the groups are shown in Table 3. The graphical representation of the primary and secondary outcomes can be seen in Figure 3 and Figure 4, respectively. There were significant within-group differences for the experimental and control groups (*p* < 0.05) in terms of the range of motion, SPADI (total, pain and disability), catastrophizing, and therapeutic alliance. There were significant within-group differences in the experimental group in terms of kinesiophobia, while there were no significant within-group differences in the control group regarding this variable.

The 2 × 2 ANOVA revealed a significant group × time interaction for the range of motion during shoulder flexion (F = 15.27; *p* < 0.001; η*p*^2^ = 0.22), SPADI-total (F = 6.28; *p* = 0.015; η*p*^2^ = 0.06) and SPADI-disability (F = 6.14; *p* = 0.01; η*p*^2^ = 0.10), but not for SPADI-pain (F = 3.94; *p* = 0.052; η*p*^2^ = 0.06). There was also a significant group × time interaction for catastrophizing (F = 8.79; *p* = 0.005; η*p*^2^ = 0.14) and kinesiophobia (F = 7.62; *p* = 0.008; η*p*^2^ = 0.12). Between the groups, the effect sizes varied from medium (0.06) to large (0.14) in favor of the experimental group.

## 4. Discussion

This study compared a PNE treatment combined with MT and EX with a physio-therapy treatment utilizing MT and EX in patients with shoulder pain. Our results indicated that the PNE program associated with MT and EX was more effective than the MT and EX program for improving the range of motion, disability, catastrophizing, and kinesiophobia but not for reducing pain. In addition, a better therapeutic alliance between the physiotherapist and patient was established in the experimental group compared to the control group. The range of motion improved in both groups, exceeding the MCID score, although there were statistically significant differences in favor of the PNE group. Overall, our results contrast with those of a previous study evaluating the efficacy of PNE as an adjunct to a physical rehabilitation protocol after arthroscopic rotator cuff repair, which showed better overall scores for active ROM in shoulder flexion, but no statistically significant differences between groups [59]. On the other hand, Louw et al. [60], who examined the application of PNE before shoulder surgery, reported similar results, where the participants showed an increase in the AROM during flexion of the affected shoulder after PNE of 5° on average, compared to 29° in our study. The difference in these results in favor of our study could be due to the fact that the patients in the Kim et al. [59] and Louw et al. [60] studies were either having shoulder surgery or awaiting surgery, which would increase kinesiophobia, leading to a decrease in movement due to the correlation between kinesiophobia and range of motion [61]. Furthermore, the study by Louw et al. [60] only included a PNE treatment, and it has been shown that PNE as a stand-alone treatment is not as effective, while PNE combined with other therapeutic treatments, especially exercises, produces significant improvements [62].

Compared to the baseline values of the SPADI scores associated with pain and disability and the total scores, both of the groups improved statistically significantly, exceeding the MCID score, and with statistically significant differences in favor of the experimental group in the total and disability score, but not in the SPADI score associated with pain. In decreasing pain and disability in shoulder pain, different studies have reported unsatisfactory results on the efficacy of PNE. [42,57,63]. The trial by Kararti et al. examined the efficacy of PNE as an adjunct to clinical outcomes in patients undergoing arthroscopic rotator cuff repair and compared the results to those from a control group treated with conventional physiotherapy [57]. The study by Myers et al. evaluated the impact of a cognitive-behavioral intervention to improve expectations towards physiotherapy and reduce the likelihood of opting for surgery [63]. Also, Ponce-Fuentes et al. investigated the effectiveness of PNE compared to Biomedical Education as part of a rehabilitation program following arthroscopic rotator cuff repair [42]. None of the trials [42,57,63] found improvements in pain and disability between the groups assessed. Moreover, in our study, no group–time interaction was observed in the SPADI-pain value, but there was an interaction with the SPADI-disability value, which may be because the reduction in the fear of movement led to early functional improvements through an increased tolerance to physical activities, before pain showed significant changes, as the final measurement was taken 5 weeks after the start of the treatment [64].

As we expected, PNE associated with MT and EX was more effective than MT and EX alone in reducing catastrophizing and kinesiophobia, with the MCD score being exceeded. Although certain studies that specifically performed PNE interventions for shoulder pain prior to or after surgery found no significant differences in these variables [42,59,60], systematic reviews of interventions that have combined PNE with exercise in chronic musculoskeletal pain have shown clinically relevant reductions in catastrophizing [65] and kinesiophobia [66]. The differences in results with the Louw et al. [60], Kim et al. [59], and Ponce-Fuentes et al. [42] studies may be due to the characteristics of the participants, since in our study the patients had not undergone shoulder surgery, nor were they awaiting this surgical intervention, and the PNE program we carried out was at group level, and not individual as in the aforementioned studies. Reduced levels of kinesiophobia may have led to increased adherence to exercise, thus improving pain intensity and disability outcomes [61].

Finally, significant differences were also found in the therapeutic alliance between the experimental and control groups. The therapeutic alliance can influence the patient’s expectations, commitment to the treatment program, motivation and self-efficacy, thus creating the ideal environment for a biomechanical exercise intervention [67].

Several limitations should be noted. Differences in the number of sessions between groups (in favor of the experimental group) may have influenced the results of this study due to factors related to the therapist–patient relationship. Even so, the treatment received by the control group is supported by clinical practice guidelines for the treatment of shoulder pain. The lack of long-term follow-up (6 and 12 months post-treatment) limits the ability to assess the sustainability of the observed benefits, making it impossible to determine whether the effects of NSP are sustained over time or whether relapses in symptoms occur. In addition, it is interesting to state that there are a restricted number of experimental trials on PNE applied specifically to chronic shoulder pain, so further research is required to obtain more definitive results and more solid conclusions on the efficacy of this technique, thus taking with caution those of the present study and their interpretation. This study has not been performed on a previous protocol, so rigorous research would also be needed to optimize training strategies and protocols to guide this approach.

Future research could explore modifications to the content of the PNE, such as the inclusion of specific modules to address psychosocial factors and comorbidities associated with shoulder pain. In addition, extending the number and duration of sessions could allow for greater depth in the key concepts of the PNE. Another aspect to consider in future research would be to consider extending the follow-up period to further assess the sustainability of the observed benefits and explore the long-term impact of the intervention. Furthermore, the findings obtained in the present study could have promising clinical implications with application in subgroups of patients, such as those with acute versus chronic shoulder pain, or in a wide repertoire of shoulder pathologies, such as adhesive capsulitis, subacromial impingement syndrome, or rotator cuff tears, a question that remains open for future research.

## 5. Conclusions

In conclusion, this trial demonstrated that PNE improved the range of motion, disability, catastrophizing, and kinesiophobia, as well as the therapeutic alliance. PNE could be an intervention option associated with MT and EX treatment in the management of shoulder pain.

## Figures and Tables

**Figure 1 brainsci-15-00125-f001:**
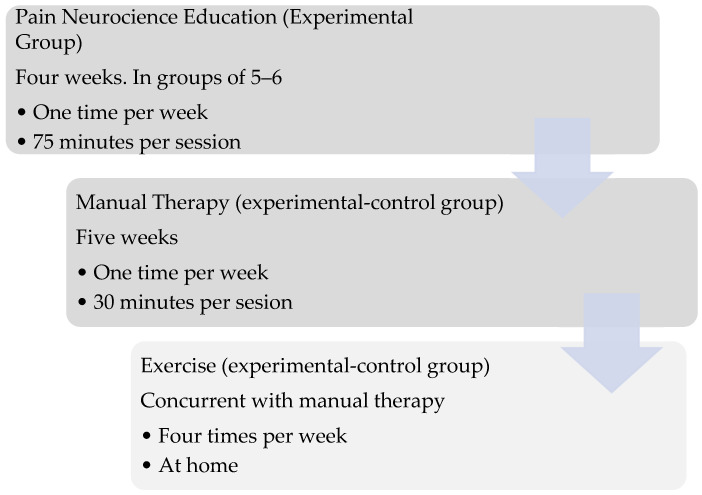
Temporal distribution of program sessions.

**Figure 2 brainsci-15-00125-f002:**
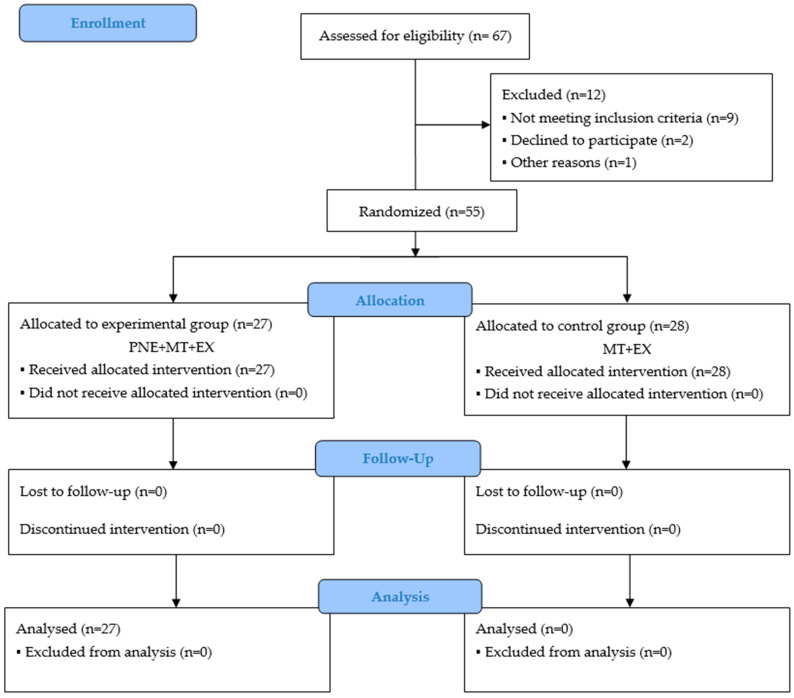
CONSORT flowchart.

**Figure 3 brainsci-15-00125-f003:**
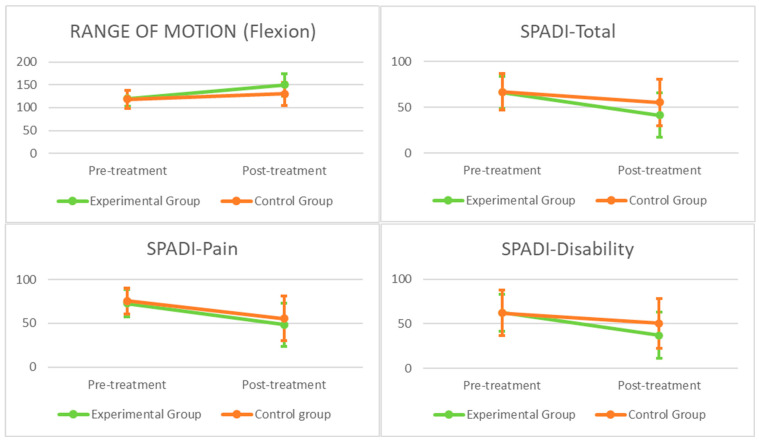
Graphical representation of primary outcomes. SPADI: Shoulder Pain and Disability Index.

**Figure 4 brainsci-15-00125-f004:**
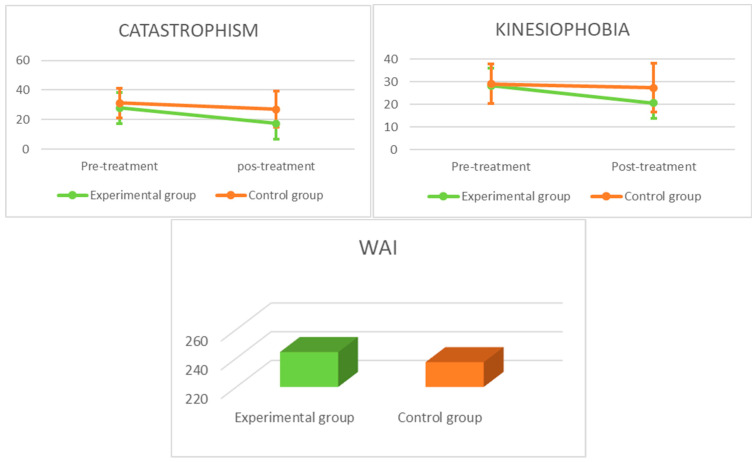
Graphical representation of secondary outcomes. WAI: Working Alliance Inventory.

**Table 1 brainsci-15-00125-t001:** Distribution of contents per session.

Session n°.	Basic Contents
Session 1	▪ Anatomy of the shoulder joint complex. ▪ Symptom—imaging relationship. ▪ Evidence on the management of shoulder pain. ▪ Epidemiology of chronic shoulder pain.
Session 2	▪ Pain as a protective system. ▪ Differences between pain and nociception. ▪ Differences Between acute and chronic pain. ▪ Sensory receptors, transduction and transmission of nociception. ▪ Pain as brain perception.
Session 3	▪ Psychosocial factors. ▪ Pain facilitation—inhibition system. ▪ Central sensitization. ▪ Kinesiophobia, catastrophizing and fear—avoidance behaviors.
Session 4	▪ Neuromatrix and neurobiology of pain. ▪ Structural changes in the brain secondary to chronic pain. ▪ Brain plasticity and reversibility of structural changes. ▪ Benefits of exercise.

**Table 2 brainsci-15-00125-t002:** Baseline characteristics of the sample.

Demographics	Experimental (N = 27)	Control (N = 28)	*p*
Age (years)	57.22 (1.57)	56.18 (1.71)	0.65
Sex (male/female)	9/18 (33.3%)	10/18 (35.7%)	0.85
Dominant side (right/left)	27/0	26/2 (92.9%)	0.15
Injured side (right/left)	19/8 (70.4%)	19/9 (67.9%)	0.84
Duration of symptoms (months)	12.33 (1.70)	13.61 (1.67)	0.59

Mean ± SD values for the continuous variables. *p*-values for differences in means between groups calculated using independent *t*-test, and χ^2^ was used for categorical variables.

**Table 3 brainsci-15-00125-t003:** Comparison of the outcome measurements between and within the groups.

		Pre-Treatment	Post-Treatment	*p*	Time	Time × Group	
Variables		Mean (SD)	Mean (SD)		F (*p*)	F (*p*)	η*p*²
ROM	E ^1,2^	120.11 (22.35)	149.85 (18.60)	0.001	79.84 (<0.001)	15.27 (<0.001)	0.22
	C ^1,2^	118.60 (25.96)	130.25 (26.67)	0.003			
SPADI -total	E ^1,2^	66.32 (17.30)	41.36 (24.25)	0.001	46.49 (<0.001)	6.28 (0.015)	0.10
	C ^2^	66.89 (19.84)	55.35 (25.45)	0.003			
SPADI -pain	E	72.88 (15.70)	48.29 (24.68)	0.001	33.86 (<0.001)	3.94 (0.052)	0.06
	C	75.50 (14.59)	55.35 (25.45)	0.008			
SPADI -disability	E	62.23 (20.71)	37.05 (25.72)	0.001	45.09 (<0.001)	6.14 (0.01)	0.10
	C	61.93 (25.53)	50.33 (28.07)	0.004			
Catastrophism	E ^1^	27.85 (10.57)	17.30 (10.53)	0.001	49.27 (<0.001)	8.79 (0.005)	0.14
	C	31.29 (10.03)	27.00 (12.14)	0.006			
Kinesiophobia	E ^1^	28.22 (7.95)	20.67 (6.72)	0.001	19.59 (<0.001)	7.62 (0.008)	0.12
	C	29.11 (8.67)	27.36 (10.70)	0.24			
WAI	E		244.30 (8.56)	0.03			
	C		237.25 (14.37)				

E: experimental group (pain neuroscience education + manual therapy + exercises); C: control group (manual therapy + exercises); SD: standard deviation; SPADI: Shoulder Pain and Disability Index. MCD: minimal detectable change. MCID: minimum clinically important difference. ^1^: exceeds MCD; ^2^: exceeds MCID. WAI: Working Alliance Inventory.

## Data Availability

The datasets used and the data analyzed in this study will be made available upon reasonable request to the corresponding author (J.A.D.-G.). The data are not publicly available due to ethical reasons.

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
