# Peer review of "Effectiveness of Adding a Pain Neuroscience Education Program to a Multimodal Physiotherapy Intervention in Patients with Chronic Shoulder Pain: A Randomized Clinical Trial"

_brainsci, 2025, doi:10.3390/brainsci15020125_

Round 1
Reviewer 1 Report
Comments and Suggestions for Authors
Dear authors,
thank you for give me the possibility to read and revise your interesting article "The Efficacy of a Pain Neuroscience Education Program in Chronic Shoulder Pain: A Randomized, Single - Blinded Trial ". Pain neuroscience education represents an emerging area of expertise for a number of professionals involved in the study of pain and could support other therapies such as manual therapy and active exercise. The following suggestions could improve the quality of your work.
INTRODUCTION
- While it introduces the concept of PNE, it could more precisely delineate the specific research gap this study aims to fill, particularly in the context of existing treatments for shoulder pain.
- The associations of manual therapy and active exercise in the treatments of shoulder pain should be support by more references, especially references concerning the most painful shoulder disease. Please take in to consideration the following articles:
*Deodato M, Martini M, Buoite Stella A, Citroni G, Ajčević M, Accardo A, Murena L. Inertial Sensors and Pressure Pain Threshold to Evaluate People with Primary Adhesive Capsulitis: Comparison with Healthy Controls and Effects of a Physiotherapy Protocol. J Funct Morphol Kinesiol. 2023 Oct 6;8(4):142. doi: 10.3390/jfmk8040142. PMID: 37873901; PMCID: PMC10594492.
*Lee JH, Jeon HG, Yoon YJ. Effects of Exercise Intervention (with and without Joint Mobilization) in Patients with Adhesive Capsulitis: A Systematic Review and Meta-Analysis. Healthcare (Basel). 2023 May 22;11(10):1504. doi: 10.3390/healthcare11101504. PMID: 37239790; PMCID: PMC10218666.
- The objective of the study should be the same described in the clinical trial registration
METHOD
- Details on the blinding process are limited. Expanding on how blinding was maintained for participants and assessors could help minimize bias and clarify the study's validity.
- the choice of mulligan Mulligan's Movement Mobilization should be justify, for example why did not used Maitland mobilization?
- table 1. It seems that you use two different font
- The choice of outcomes measures should be better justify with references, for example, you used goniometer but why not inertial sensor?
- A longer follow-up period would be beneficial to assess the sustainability of the intervention's effects over time
DISCUSSION
- The discussion would benefit from more detailed suggestions for future research, including potential modifications to the PNE and the exploration of long-term effects through extended follow-up periods.
- A more comprehensive discussion of the study's limitations would provide a more balanced view.
- More thorough consideration of the study's generalizability to different shoulder pain populations and settings would help readers assess the findings' applicability.
Author Response
Title: The Efficacy of a Pain Neuroscience Education Program in Chronic Shoulder Pain: A Randomized, Single - Blinded Trial
Response to Reviewer 1
|
||
|
||
Thank you very much for taking the time to review this manuscript. Please find the detailed responses below and the corresponding corrections highlighted changes in the re-submitted files.
|
||
Comments 1: While it introduces the concept of PNE, it could more precisely delineate the specific research gap this study aims to fill, particularly in the context of existing treatments for shoulder pain.
|
||
Response 1: Thank you for pointing this out. We agree with this comment. Therefore, we have added the requested information highlighted in red in the manuscript on page number 2, lines 74-82.
|
||
Comments 2: The associations of manual therapy and active exercise in the treatments of shoulder pain should be support by more references, especially references concerning the most painful shoulder disease. Please take in to consideration the following articles:
*Deodato M, Martini M, Buoite Stella A, Citroni G, Ajčević M, Accardo A, Murena L. Inertial Sensors and Pressure Pain Threshold to Evaluate People with Primary Adhesive Capsulitis: Comparison with Healthy Controls and Effects of a Physiotherapy Protocol. J Funct Morphol Kinesiol. 2023 Oct 6;8(4):142. doi: 10.3390/jfmk8040142. PMID: 37873901; PMCID: PMC10594492.
*Lee JH, Jeon HG, Yoon YJ. Effects of Exercise Intervention (with and without Joint Mobilization) in Patients with Adhesive Capsulitis: A Systematic Review and Meta-Analysis. Healthcare (Basel). 2023 May 22;11(10):1504. doi: 10.3390/healthcare11101504. PMID: 37239790; PMCID: PMC10218666. |
||
Response 2: Agree. We have, accordingly, modified the text and added the references indicated, as well as others necessary to include the suggestions for change made to emphasize this point. Therefore, we have added the requested information highlighted in red in the manuscript on page number 2, lines 57-62.
|
||
Comments 3: The objective of the study should be the same described in the clinical trial registration.
|
Response 3: Agree. We have, accordingly, modified the objectives as described in the clinical trial registration.
Comments 4: Details on the blinding process are limited. Expanding on how blinding was maintained for participants and assessors could help minimize bias and clarify the study's validity.
Response 4: Thank you for pointing this out. We agree with this comment. Therefore, we have expanded the details on blinding of participants and evaluators. We have added the requested information highlighted in red in the manuscript on page number 3, lines 118-121 and lines 128-130.
Comments 5: The choice of mulligan Mulligan's Movement Mobilization should be justify, for example why did not used Maitland mobilization?
Response 5: Agree. We have justified the choice of Mulligan's technique on the basis of the available evidence and the preference and knowledge of the therapist who performed the intervention. We have added the requested information highlighted in red in the manuscript on page number 4, lines 142-146.
Comments 6: Table 1. It seems that you use two different font.
Response 6: Thank you for pointing this out. Table 1 has already been modified.
Comments 7: The choice of outcomes measures should be better justify with references, for example, you used goniometer but why not inertial sensor?
Response 7: Thank you for pointing this out. We have expanded the references of goniometry and justified its use. The main reasons for the use of the goniometer were not only because the examiner was familiar with its use, but also because of the limited number of existing validation studies of the inertial sensor. We have added the requested information highlighted in red in the manuscript on page number 5, lines 196-199.
Comments 8: A longer follow-up period would be beneficial to assess the sustainability of the intervention's effects over time.
Response 8: Agree. In our study, we opted for a 5-week follow-up period due to project timelines and the limited duration of the research grant, which only allowed us to evaluate the immediate and short-term outcomes of the intervention. This has been acknowledged as a limitation of the study and is included as a proposal for future research.
Comments 9: The discussion would benefit from more detailed suggestions for future research, including potential modifications to the PNE and the exploration of long-term effects through extended follow-up periods.
Response 9: Thank you for pointing this out. We agree with this comment. Therefore, we have added the requested information highlighted in red in the manuscript on page number 10, lines 388-391.
Comments 10: A more comprehensive discussion of the study's limitations would provide a more balanced view.
Response 10: Agree. We have added the requested information highlighted in red in the manuscript on page number 10, lines 375-384.
Comments 11: More thorough consideration of the study's generalizability to different shoulder pain populations and settings would help readers assess the findings' applicability.
Response 11: Thank you for pointing this out. We agree with this comment. Therefore, we have added the requested information highlighted in red in the manuscript on page number 10, lines 391-395.

Reviewer 2 Report
Comments and Suggestions for Authors
Thanks to the author for conducting this study.
In the discussion, the author used vocabularies like “better than superior to” other studies. Please change this to be more scientific argument.
please highlight what is the gap in research that is covered by the results of this study.
what was the level of education of the participants, are equally distributed among the study groups?
please present the most significant results in a graph to attract the attention of the readers.
Author Response
Title: The Efficacy of a Pain Neuroscience Education Program in Chronic Shoulder Pain: A Randomized, Single - Blinded Trial
Response to Reviewer 2
|
||
|
|
|
Thank you very much for taking the time to review this manuscript. Please find the detailed responses below and the corresponding corrections highlighted changes in the re-submitted files.
|
||
Comments 1: In the discussion, the author used vocabularies like “better than superior to” other studies. Please change this to be more scientific argument.
|
||
Response 1: Agree. We have therefore made the requested modification highlighted in red in the manuscript on page 10, lines 321,324-325.
Comments 2: Please highlight what is the gap in research that is covered by the results of this study.
Response 2: Thank you for pointing this out. We agree with this comment. Therefore, we have added the requested information highlighted in red in the manuscript on page number 2, lines 74-82.
Comments 3: What was the level of education of the participants, are equally distributed among the study groups?
Response 3: Thank you for pointing this out. The educational level of the participants was not studied as it was not considered relevant, as only one of the groups received pain neuroscience education, and by assessing the study by Bilterys et al. (doi: 10.1016/j.msksp.2021.102494) who reported that there were no significant clinical differences in the efficacy of pain neuroscience education in terms of level of education. To ensure homogeneity of the groups, pre-randomization was performed.
|
Comments 4: Please present the most significant results in a graph to attract the attention of the readers.
Response 4: Agree. We have made graphical representations of the primary and secondary results in inter-group comparisons. Page number 9.

Round 2
Reviewer 1 Report
Comments and Suggestions for Authors
I thank authors for the work done
Reviewer 2 Report
Comments and Suggestions for Authors
Thanks!